# Adaptive Sparse Multi-Block PLS Discriminant Analysis: An Integrative Method for Identifying Key Biomarkers from Multi-Omics Data

**DOI:** 10.3390/genes14050961

**Published:** 2023-04-23

**Authors:** Runzhi Zhang, Susmita Datta

**Affiliations:** Department of Biostatistics, University of Florida, Gainesville, FL 32603, USA; runzhi.zhang@ufl.edu

**Keywords:** data integration, multi-omics, asmbPLS-DA, classification

## Abstract

With the growing use of high-throughput technologies, multi-omics data containing various types of high-dimensional omics data is increasingly being generated to explore the association between the molecular mechanism of the host and diseases. In this study, we present an adaptive sparse multi-block partial least square discriminant analysis (asmbPLS-DA), an extension of our previous work, asmbPLS. This integrative approach identifies the most relevant features across different types of omics data while discriminating multiple disease outcome groups. We used simulation data with various scenarios and a real dataset from the TCGA project to demonstrate that asmbPLS-DA can identify key biomarkers from each type of omics data with better biological relevance than existing competitive methods. Moreover, asmbPLS-DA showed comparable performance in the classification of subjects in terms of disease status or phenotypes using integrated multi-omics molecular profiles, especially when combined with other classification algorithms, such as linear discriminant analysis and random forest. We have made the R package called *asmbPLS* that implements this method publicly available on GitHub. Overall, asmbPLS-DA achieved competitive performance in terms of feature selection and classification. We believe that asmbPLS-DA can be a valuable tool for multi-omics research.

## 1. Introduction

The improvements in high-throughput technology have enabled the collection of different types of omics data such as transcriptomics, epigenomics, proteomics, and metabolomics quickly and cost-effectively for the same subject. Each type of omics data, on its own, may contribute unique information. By identifying biomarkers within each type of omics data, we can gain insight as to which biological pathways or processes are different between different phenotypic groups. However, when analyzing only one type of omics data, the correlation between different omics layers is not considered, preventing us from understanding the complex relationship (molecular mechanism) between the multi-omics data and the host systematically and holistically. Therefore, in order to draw a more comprehensive view of systemic biological processes, omics data from the different platforms have to be integrated and analyzed [1]. 

Multi-omics data are often structured in blocks and characterized by hundreds or thousands of variables with only a small sample size. Meanwhile, they often imply a high degree of correlation within and between different types of omics data. With a large number of variables, the interpretation of their collective contribution to the process can be a hard task. Therefore, there is a need to perform variable selection to obtain the most relevant and predictive features for each block.

In the past decade, many approaches have been proposed for identifying predictive biomarkers that are relevant to the patient’s phenotypes. For some methods, such as L1-regularized logistic regression, L1-regularized multinomial logistic regression [2], and sparse partial least square discriminant analysis (sPLS-DA) [3], the group structure of the data is not considered. When applied to multi-omics data, different types of data need to be merged at first and the source of the variables is ignored, which could lead to the loss of information for specific omics data. For approaches considering the group structure, integrative Lasso with penalty factors (IPF-Lasso) [4] is an extension of the standard Lasso [5], which applies different penalty factors to different blocks. However, one limitation of the Lasso-based method is that it tends to select only one variable from a group of highly correlated variables, resulting in less relevant features selected in such scenarios [6]. In addition, Data Integration Analysis for Biomarker discovery using Latent cOmponents (DIABLO) [7] is another newly developed multi-omics integrative method for finding the biomarkers, which extends sparse generalized canonical correlation analysis (sGCCA) [8], a multivariate dimension reduction technique that uses singular value decomposition and selects co-expressed variables from several omics datasets to a supervised framework.

In this study, we present adaptive sparse multi-block partial least square discriminant analysis (asmbPLS-DA), an extension of our previously proposed method, asmbPLS [9], which is designed for categorical outcome instead of continuous outcome, such as disease status (e.g., case/control, severe/moderate/mild). This method aims to identify predictive biomarkers from multi-omics data while simultaneously discriminating between different phenotypic groups, taking into consideration the group structure. To accomplish this, asmbPLS-DA maximizes the covariance between two latent variables, which represent the information from predictors and outcome matrices, respectively. Different decision rules are used to determine the group to which each sample belongs. While the binary outcome is the most commonly considered scenario, asmbPLS-DA also allows for multiclass outcomes with more than two groups, where the PLS2 structure is utilized, and each column of the outcome matrix represents a different group or class. Each sample is then assigned to belong to a specific group by indicating 1 or 0 in the respective column of the outcome matrix.

Our study demonstrates that asmbPLS-DA performs well in identifying predictive biomarkers and achieves comparable classification performance for both binary and multiclass outcomes. We validated the method using both simulated and real data. To facilitate its use, we have developed an R package called *asmbPLS*, which is publicly available on GitHub (https://github.com/RunzhiZ/asmbPLS). The package also includes visualization functions that can assist with result interpretation.

## 2. Materials and Methods

In this section, we introduce adaptive sparse partial least square discriminant analysis (asmbPLS-DA), which is based on adaptive sparse partial least square (asmbPLS) [9] for discriminant analysis. Different decision rules were included for the purpose of classification. We let the matrix including B blocks (different omics data) $X=\left[X^{1},\;X^{2},\ldots ,\;X^{B}\right]$ and the matrix Y be our predictor matrix and outcome matrix, respectively, on the same n samples. The samples are split into G groups of sizes $n_{1}+n_{2}+\cdots +n_{G}=n$. For Y, it is coded as dummy variables (1/0), with only one column for the binary outcome (G = 2), and G columns for the multiclass outcome (G≥3).

### 2.1. Introduction to asmbPLS

#### 2.1.1. Model Building

In asmbPLS, the goal is to construct a set of orthogonal PLS components that maximize the covariance between the two latent variables $t_{j}^{super}\mathrm{and}\;u_{j}$, which represent information from X and Y, respectively. X and Y are always centered and scaled. For jth PLS component (j = 1, 2, …, J), the dimension reduction is conducted by taking a linear combination of the predictor features to obtain the block score $t_{j}^{b}=\frac{X^{b}\omega_{j}^{b}}{{\left(m_{X^{b}}\right)}^{\frac{1}{2}}}$ (b=1,…, B) for each block, where $\omega_{j}^{b}$ contains the block variable weights in $X^{b}$**,** indicating the relevance of variables, and $m_{X_{b}}$ is the number of variables in $X^{b}$ for block scaling. Within each block, the soft thresholding function $sparse\left(x,\;\lambda \right)=sign\left(x\right){\left(\left|x\right|-\lambda \right)}_{+}$ is implemented on $\omega_{j}^{b}$ with the specific quantile of weights used as λ for feature selection. By doing this, the most relevant features will be retained given pre-defined quantiles for each block. After calculating $t_{j}^{b}$ for each block, all block scores are combined into a new matrix $T_{j}$. The dimension reduction is implemented again by taking a linear combination of the different block scores $t_{j}^{super}=T_{j}\omega_{j}^{super}$, where $\omega_{j}^{super}$ is the weight indicating the relevance of each block and $t_{j}^{super}$ is the super score representing the information of predictor matrix X. Similarly, $u_{j}$ is a summary vector of Y, i.e., $u_{j}=Yq_{j}$ ($q_{j}$ being the weight for each column of Y).

The problem can be formally expressed as follows:(1)$$\max cov\left(t_{j}^{super},\;u_{j}\right)with\;t_{j}^{b}=\frac{X^{b}\omega_{j}^{b}}{{\left(m_{X^{b}}\right)}^{\frac{1}{2}}},\;T_{j}=\left[t_{j}^{1},\;t_{j}^{2},\ldots ,t_{j}^{B}\right],\;u_{j}=Yq_{j},\;and\;t_{j}^{super}\;=T_{j}\omega_{j}^{super},\;subject\;to\;\left|\left|\omega_{j}^{b}\right|\right|=\left|\left|\omega_{j}^{super}\right|\right|=1$$

Once all the parameters are calculated for the first PLS component, ***X*** and ***Y*** are deflated, and then the deflated ***X*** and ***Y*** are used for the calculation of the second PLS component, and so on. 

#### 2.1.2. Prediction

Once the asmbPLS model is built, we can then predict the outcome for the original and the new samples. Specifically, given the new data $X^{new}$, we can calculate $t_{j}^{b}$, $T_{j}$, and $t_{j}^{super}$. Therefore, the outcome for the unknown sample can be predicted by:(2)$$Y^{predict}=t_{1}^{super}q_{1}^{⊤}+t_{2}^{super}q_{2}^{⊤}+t_{3}^{super}q_{3}^{⊤}+\cdots$$
where the prediction can depend on the number of PLS components used.

### 2.2. asmbPLS Extended to asmbPLS-DA

For asmbPLS-DA, we focus on the categorical outcome instead of the continuous outcome. The extension of asmbPLS to asmbPLS-DA is straightforward. For the binary outcome, a matrix with one column dummy variable is used as the outcome matrix (n×1); for the multiclass outcome, a matrix with G columns dummy variables is used (n×G). 

#### 2.2.1. Data Pre-Processing

In asmbPLS-DA, the pre-processing of data is different. X is always column-wise centered and scaled; Y is only centered. Usually, for the binary outcome, 0.5 (the average of two group indicators, i.e., 0 and 1) is used as the cutoff for the group decision when we have equal sample sizes in two groups. However, when the group sizes are unequal, the boundary is shifted toward the larger group, resulting in more misclassification. Therefore, to handle this, we implement the weighted center in asmbPLS-DA by subtracting the average of the means of the two groups, A and B, i.e., $\left({\bar{X}}_{A}+{\bar{X}}_{B}\right)/2$, for each feature in X. We conduct the same weighted center for Y using the same group information. In addition, for the multiclass outcome, a similar procedure is applied. For example, if we have three groups (A, B, and C), then we use the average of the means of the three groups $\left({\bar{X}}_{A}+{\bar{X}}_{B}+{\bar{X}}_{C}\right)/3$ to implement the weighted center.

#### 2.2.2. Decision Rules

Over the last two decades, different decision rules have been proposed to translate Y-estimate into the meaningful group. We have included many of them in the proposed algorithm. The details are listed in Table 1. 

Among the included decisions, the Euclidean distance (ED) is one of the most used measures, which is defined as:(3)$$d_{ig}^{2}=\left(X_{i}-{\bar{X}}_{g}\right){\left(X_{i}-{\bar{X}}_{g}\right)}^{⊤}$$
where $d_{ig}$ is the ED of sample i to the centroid of group g (${\bar{X}}_{g}$). The Mahalanobis distance (MD) [10] is another popular measure, which seeks to measure the correlation between variables:(4)$$d_{ig}^{2}=\left(X_{i}-{\bar{X}}_{g}\right)S^{-1}{\left(X_{i}-{\bar{X}}_{g}\right)}^{⊤}$$
(5)$$S=\frac{\left(n_{1}-1\right)S_{1}+\cdots +\left(n_{g}-1\right)S_{g}+\cdots }{n_{1}+\cdots n_{g}+\cdots +n_{G}-G}$$
where $S_{g}$ is the variance–covariance matrix for group g.

#### 2.2.3. Parameter Tuning and Model Selection

We let B = 2 and the number of PLS components = 3, for instance. K-fold CV is used to tune the quantile combination for different PLS components in different blocks to obtain the best quantile combination giving the highest balanced accuracy (BA). For the binary outcome, BA is defined as (sensitivity + specificity)/2, where sensitivity = true positive/(true positive + false negative) and specificity = true negative/(true negative + false positive); for the multiclass outcome, BA is defined as average recall or sensitivity across all the groups. Tuning these parameters is equivalent to choosing the “degree of sparsity”, i.e., the number of non-zero weights for each PLS component in each block. 

In the K-fold CV, the samples are randomly placed into K roughly equal groups with the ratios of samples from different classes being equal in all the groups. This step is repeated $N_{CV}$ times to generate $N_{CV}$ different sets of groups. For each set with K groups, the samples from the kth group are used as the validation set, while the remaining samples serve as the training set, with this process repeated for k=1,…, K. For convenience, the same set of combinations of the degree of sparsity, i.e., $quantile^{block_{1}}=\left\{q_{1}^{block_{1}},\ldots ,q_{a}^{block_{1}}\;\right\}$ and $quantile^{block_{2}}=\left\{q_{1}^{block_{2}},\ldots ,q_{b}^{block_{2}}\;\right\}$ are used for different PLS components. The following steps are then taken: (1) asmbPLS-DA models using the first l (l starts from 1) PLS component(s) with different quantile combinations used for the lth PLS component are fitted based on the training set; (2) BA for each model is then calculated based on the validation set; (3) the average K-fold BA across $N_{CV}$ sets is calculated for each combination: (6)$${\mathrm{BA}}_{N_{\mathrm{CV}},K}=\frac{1}{N_{CV}}\overset{N_{CV}}{\underset{n_{CV}=1}{\sum }}\frac{1}{K}\overset{K}{\underset{k=1}{\sum }}BA_{n_{CV},k}$$
where $BA_{n_{CV},k}$ is the corresponding BA using samples from the kth group of $n_{CV}$th set as the validation; (4) the combination with the lowest ${\mathrm{BA}}_{N_{\mathrm{CV}},K}$ is chosen for lth PLS component; (5) the data deflation is then implemented for X and Y, and we let l=l+1; and (6) steps (1)–(5) are repeated until we obtain the combinations for all the 3 PLS components. Note that the combination with the lowest ${\mathrm{BA}}_{N_{CV},K}$ obtained from the previous steps will be used for the first (*l*–1)th PLS component(s).

In our study, we choose K and $N_{cv}$ to be 5 and 10, respectively, which is enough for parameter tuning. The selection of the quantiles for each block is usually based on prior knowledge of the corresponding omics data. The blocks with only a small number of relevant features should be given higher quantiles and vice versa. In addition, if there is no prior knowledge, a wider range of quantiles should be tried.

Using the results of the CV, we can determine the number of PLS components used for classification as well. To avoid the over-fitting issue, the strategy for selecting the optimal number of PLS components is: (1) let the initial number of components be comp = 1; (2) check whether including one more component decreases the BA by 0.005, i.e., $BA_{comp}+0.005\leq BA_{comp+1}$; and (3) if so, let comp = comp + 1 and go back to step (2); otherwise, let comp be the selected number of components.

#### 2.2.4. Vote Algorithm

In addition to the different decision rules, we have included several vote algorithms to help the discrimination. (1) The unweighted vote, where each decision rule used will be assigned equal weight, and the final classification is based on the results of the methods. (2) The weighted vote, where each decision rule is assigned a weight based on the BA with the optimal quantile combination, i.e., $weight_{i}=\log\left(\frac{BA_{i}}{1-BA_{i}}\right)$. For $BA_{i}\leq 0.5$, $weight_{i}$ is set to 0. (3) The ranked vote, in addition to BA, the overall accuracy, recall, precision, and F1 score for the optimal quantile combination has also been collected. The rank of the decision rule is calculated for different measures and a combined rank is obtained. The decision rule with the highest combined rank is used for the final classification.

#### 2.2.5. Another Classification Strategy

In addition to the above-mentioned original asmbPLS-DA classification algorithms, we have combined asmbPLS-DA with other methods for classification. Once the asmbPLS-DA model is built, the super score of different PLS components or the selected features can be used as predictors for other classification algorithms, such as RF and LDA. By doing this, we aim to test if this two-stage strategy will increase the classification capability of asmbPLS-DA.

### 2.3. Technicalities and Implementations

All the implementations were conducted using R 4.2.1 [11] in the “HiperGator 3.0” high-performance computing cluster, which includes 70,320 cores with 8 GB of RAM on average for each core, at the University of Florida. 

The real data downloading and pre-processing were implemented using the R package *TCGAbiolinks* (2.25.3) [12]. The *GDCquery* function was used to query the required GDC data, where the project option was set to be “TCGA-BRCA”. For gene data, we set data.category = “Transcriptome Profiling”, data.type = “Gene Expression Quantification”, and workflow.type = “STAR-Counts”; for miRNA data, we set data.category = “Transcriptome Profiling”, data.type = “miRNA Expression Quantification”, and workflow.type = “BCGSC miRNA Profiling”; for protein data, we set data.category = “Proteome Profiling” and data.type = “Protein Expression Quantification”.

The R package *sfsmisc* (1.1-14) [13] was used to make the variance–covariance matrix in the simulation studies positive definite. L1-regularized logistic regression and L1-regularized multinomial logistic regression were implemented using R package *glmnet* (4.1-7) [14]; sPLS-DA and DIABLO were implemented using R package *mixOmics* (6.22.0) [15]; and IPF-Lasso was implemented using R package *ipflasso* (1.1) [4]. RF was implemented using R package *randomForest* (4.7-1.1) [16]. LDA was implemented using R package *MASS* (7.3-58.3) [17]. The use of R packages followed the suggestion from the corresponding package’s tutorial. 

## 3. Results and Discussion

asmbPLS-DA employs different decision rules and vote functions that can impact its performance in various scenarios. Thus, the identification of the best decision rules and vote functions is crucial for selecting the optimal model. Moreover, asmbPLS-DA can be combined with other classification methods, such as random forest (RF) [18] and linear discriminant analysis (LDA) [19], to improve the accuracy and reduce the risk of overfitting. In this case, the outputs of asmbPLS-DA, such as the super score (of the first two or three PLS components) or selected features, can be utilized as predictors for the other classification algorithms. 

### 3.1. Simulation Strategies

The objective of the simulation studies is to investigate the performance of asmbPLS-DA and compare it with other classification methods. To accomplish this, we consider two correlated high-dimensional predictor blocks that vary in the total number of variables and the number of highly correlated variables in each block. In addition, we examine two phenotypic group scenarios, including binary and multiclass (three groups) outcomes. By conducting simulation studies with varying parameter settings, we aim to comprehensively evaluate the performance of asmbPLS-DA and other methods for feature selection and classification. 

We simulate *n* samples, *q* variables in the first block $X^{1}$, and *p* variables in the second block $X^{2}$. Among the *q + p* variables, $q_{h}$ variables from $X^{1}$ and $p_{h}$ variables from $X^{2}$ are simulated to be highly correlated, and we set the correlation between all other variables to be from low to moderate. The highly correlated part and the non-highly correlated part are simulated from the multivariate normal distribution, respectively:$$X_{1}^{1},\ldots ,X_{q_{h}}^{1},\;X_{1}^{2},\ldots ,\;X_{p_{h}}^{2}∼MN\left(\mu_{h},\;\sum_{h}\right)$$
$$X_{q_{h}+1}^{1},\;\ldots ,\;X_{q}^{1},\;X_{p_{h}+1}^{2},\;\ldots ,\;X_{p}^{2}∼MN\left(\mu_{l},\;\sum_{l}\right)$$
where mean (variance) for variables in $X^{1}$ and $X^{2}$ are set to 0 (1) and 5 (3), respectively. To simulate the within-block and between-block correlation, for $\sum_{h}$ and the variance–covariance matrix for the highly correlated part, we set all the non-diagonal elements to be $c_{ij}\sigma_{i}\sigma_{j}$, where $\sigma_{i}$ and $\sigma_{j}$ are the standard deviations for *i*th and *j*th features, respectively, and $c_{ij}∼Uniform\left(0.6,\;0.9\right)$. Similarly, we did the same for $\sum_{l}$ with $c_{ij}∼Uniform\left(-0.5,\;0.5\right)$. We used the *posdefify* function from the R package *sfsmisc* to make the variance–covariance matrix positive definite. n, q, and $q_{h}$ are always set to be 100, 1000, and 100 while we vary the number of p and $p_{h}$ in different scenarios, including (1) p = 50 and $p_{h}$ = 5, where there is only a small number of features in $X^{2}$; (2) p = 200 and $p_{h}$ = 20, where there are more features in $X^{2}$ but still much fewer than in $X^{1}$; and (3) p = 1000 and $p_{h}$ = 100, where $X^{1}$ and $X^{2}$ have an equal number of features.

Once $X^{1}$ and $X^{2}$ are simulated, they are centered and scaled ($X^{1.scale}$ and $X^{2.scale}$) and then used to simulate the binary and multiclass outcomes Y following the idea of logistic regression and multinomial logistic regression. Only the first ten variables in $X^{1}$ and the first five variables in $X^{2}$ are set to be relevant to the outcome with non-zero coefficients, while the coefficients of all the other variables are equal to zero.

For binary outcome:$${\left(\beta^{1}\right)}^{⊤}=\left(\beta_{1}^{1},\ldots ,\;\beta_{10}^{1},\;0,\;\ldots ,\;0\right)\;{\left(\beta^{2}\right)}^{⊤}=\left(\beta_{1}^{2},\ldots ,\;\beta_{5}^{2},\;0,\;\ldots ,\;0\right)\;\beta_{nonzero}^{⊤}=\left(\beta_{1}^{1},\ldots ,\;\beta_{10}^{1},\;\beta_{1}^{2},\ldots ,\;\beta_{5}^{2}\right)\;\beta_{nonzero}∼MN\left(0_{15},\;5I_{15}\right)\;\eta =X^{1.scale}\beta^{1}+X^{2.scale}\beta^{2}+r\times e\;e∼MN\left(0_{n},5I_{n}\right)\;Prob=\frac{1}{1+\exp\left(-\eta \right)}\;Y_{i}∼Binomial\left(1\;,\;Prob_{i}\right)\;\left(i=1,\ldots ,\;n\right)\;Y_{n\times 1}^{⊤}=\left(Y_{1},\ldots ,\;Y_{n}\right)$$
where I is the identity matrix and r is used to indicate different scale of noise.

For multiclass outcome:$${\left(\beta^{class1}\right)}^{⊤}=(0_{p}^{⊤},\;0_{q}^{⊤})\;{\left(\beta^{class2}\right)}^{⊤}=\left({\left(\beta^{1.2}\right)}^{⊤},\;{\left(\beta^{2.2}\right)}^{⊤}\right)\;{\left(\beta^{class3}\right)}^{⊤}=\left({\left(\beta^{1.3}\right)}^{⊤},\;{\left(\beta^{2.3}\right)}^{⊤}\right)\;Prob^{class1}=\exp\left(X^{1.scale}0_{p}+X^{2.scale}0_{q}\right)\;Prob^{class2}=\;\exp\left(X^{1.scale}\beta^{1.2}+X^{2.scale}\beta^{2.2}+r\times e_{2}\right)\;Prob^{class3}=\exp\left(X^{1.scale}\beta^{1.3}+X^{2.scale}\beta^{2.3}+r\times e_{3}\right)\;e_{2}∼MN\left(0_{n},5I_{n}\right)\;e_{3}∼MN\left(0_{n},5I_{n}\right)\;Prob=\left(Prob^{class1},\;Prob^{class2},Prob^{class3}\right)\;Y_{i}∼Multinomial\left(1\;,\;Prob_{i}\right)\;\left(i=1,\ldots ,\;n\right)\;Y_{n\times 3}^{⊤}=\left(Y_{1}^{⊤},\ldots ,\;Y_{n}^{⊤}\right)$$
where $\beta^{1.2}$ and $\beta^{1.3}$ have the same settings as $\beta^{1}$, and $\beta^{2.2}$ and $\beta^{2.3}$ have the same settings as $\beta^{2}$.

Since only the first ten variables in $X^{1}$ and the first five variables in $X^{2}$ are set to be relevant, we consider four different variable structures for each setting mentioned above by replacing the order of the variables in different blocks. (1) cor: among the relevant variables, all of them are highly correlated variables; (2) order: among the relevant variables, only the first six variables in $X^{1}$ and the first three variables in $X^{2}$ are highly correlated variables; (3) random: the order of all the variables is random; and (4) inverse: none of the relevant variables are highly correlated variables. In addition, the r is set to be different values, including 0, 1, 2, 3, and 5, to indicate different scales of noise. Therefore, we have 60 scenarios for each type of outcome. For each scenario, 100 datasets are generated, and additional $n_{test}=n$ samples are generated using the same design parameters to serve as the test set for each dataset.

### 3.2. Simulation Results

To evaluate the performance of asmbPLS-DA and compare it with other classification methods, we conducted a comprehensive analysis of feature selection, classification accuracy, and computation efficiency. We compared asmbPLS-DA with L1-regularized logistic regression and L1-regularized multinomial logistic regression for binary and multiclass outcomes, respectively. In addition, we examined sPLS-DA and DIABLO for both binary and multiclass outcomes and IPF-Lasso for binary outcomes. To perform the comparisons, we used 5-fold cross-validation (CV) with 10 repetitions for asmbPLS-DA, sPLS-DA, IPF-Lasso, and DIABLO. We used the corresponding R package name *glmnet* to represent L1-regularized logistic regression and L1-regularized multinomial logistic regression in the figures for binary and multiclass outcomes, respectively. For asmbPLS-DA, we conducted the CV with varying quantile combinations for each scenario. In scenarios with p = 50, we used $quantile^{block_{1}}$ = {0.975, 0.98, 0.985, 0.99, 0.995} and $quantile^{block_{2}}$ = {0.7, 0.9, 0.9}; in scenarios with p = 200, we used $quantile^{block_{1}}$ = {0.975, 0.98, 0.985, 0.99, 0.995} and $quantile^{block_{2}}$ = {0.925, 0.95, 0.975}; and in scenarios with p = 1000, we used $quantile^{block_{1}}$ = {0.975, 0.98, 0.985, 0.99, 0.995} and $quantile^{block_{2}}$ = {0.975, 0.98, 0.985, 0.99, 0.995}.

#### 3.2.1. Feature Selection

In our simulation setting, only the first ten variables in $X^{1}$ and the first five variables in $X^{2}$ are set to be relevant. The sensitivity and specificity for each block were calculated based on the feature selection results. 

Figure 1 displays the overall sensitivity and specificity of different methods for the binary outcome. When the relevant features are highly correlated (structure = cor), DIABLO exhibits nearly the highest sensitivity in scenarios with p = 50 and 200, while L1-regularized logistic regression and IPF-Lasso show the lowest sensitivity in a different dimension setting due to the Lasso-based method’s limitation in selecting only one variable from a group of highly correlated variables. asmbPLS-DA performs similarly to DIABLO in scenarios with p = 50, 200, and low noise, and performs better than all the other methods in scenarios with p = 1000 and low noise. However, in scenarios with structures = order, random, and inverse, DIABLO’s performance is nearly the worst. Among these scenarios with p = 50, asmbPLS-DA shows higher sensitivity than all the other methods. As p increases to 200 and 1000, the performance of asmbPLS-DA is similar to other methods, except for DIABLO. Correspondingly, higher sensitivity is accompanied by lower specificity. Among all methods, sPLS-DA has the lowest specificity. More details of the sensitivity and specificity of different blocks for the binary outcome are presented in Appendix A. In scenarios with p = 50, asmbPLS-DA and DIABLO display higher sensitivity than methods that do not consider group structure for the smaller block $X^{2}$, as methods without group information may assign more weight to the block with more features. 

For the multiclass outcome, the results of feature selection are presented in Figure 2. In nearly all the scenarios except structure = cor and p = 200, asmbPLS-DA performs the best or close to the best regarding sensitivity. In addition, the specificity of asmbPLS is the second-best among all the methods. Appendix A shows the corresponding sensitivity and specificity of different blocks, and a similar trend in Appendix A can be found.

#### 3.2.2. Classification Performance

The classification performance of all the methods was measured in terms of overall accuracy with models fitted on the training set and then applied to the test set.

For the binary outcome, Appendix A compares the different asmbPLS-DA methods. In scenarios with structure = cor and p = 50, 200, RF using features selected by asmbPLS-DA performs the best, followed by the original asmbPLS-DA methods with different decision rules and vote functions. The performance of asmbPLS-DA with LDA is good in low noise but decreases rapidly with increased noise, as seen in scenarios with structure = random and inverse and p = 50, 200. Furthermore, LDA and RF using the super score of asmbPLS-DA show relatively lower accuracy than other asmbPLS-DA methods in most scenarios. For scenarios with structure = order, the original asmbPLS-DA methods show the best performance overall. Additionally, for p = 1000, the original asmbPLS-DA performs the best. 

Based on the comparison among the asmbPLS-DA methods, we have selected a subset of asmbPLS-DA methods to compare with other methods for simplicity (Figure 3). The selected methods are asmbPLS-DA using fixed cutoff, RF, and LDA using features selected by asmbPLS-DA. In scenarios with structure = cor, sPLS-DA and DIABLO perform the best, followed by L1-regularized logistic regression and IPF-Lasso. RF using features selected by asmbPLS-DA gradually outperforms L1-regularized logistic regression and IPF-Lasso with increased noise. In scenarios with structure = order, random, and inverse, asmbPLS-DA methods perform worse than sPLS-DA, IPF-Lasso, and L1-regularized logistic regression with low noise, but gradually improve with increased noise and show higher or similar accuracy compared to other methods. DIABLO performs the worst in these scenarios. 

For the multiclass outcome, the comparison among all asmbPLS-DA methods shows that RF using features selected by asmbPLS-DA performs the best in scenarios with structure = cor (Appendix A). For structure = order, random, and inverse, LDA using the super scores shows overall better performance than the other asmbPLS-DA methods. In addition, the performance of asmbPLS-DA with Max Y is good in scenarios with p = 1000. Similarly, we select RF using features selected by asmbPLS-DA, LDA using super scores of asmbPLS-DA and asmbPLS-DA with Max Y for comparison with other methods for simplicity.

According to Figure 4, asmbPLS-DA methods generally perform worse than L1-regularized multinomial logistic regression and sPLS-DA, except for RF using asmbPLS-DA selected features, which show close-to-best performance in scenarios with structure = cor.

#### 3.2.3. Computation Efficiency

Table 2 displays the computation time for each method in different settings. For the binary outcome, L1-regularized logistic regression is the fastest, followed by IPF-Lasso, asmbPLS-DA, and then sPLS-DA. For the multiclass outcome, L1-regularized multinomial logistic regression is the fastest, followed by sPLS-DA and asmbPLS-DA. DIABLO is the slowest among all the methods, which is not implemented for real data applications.

### 3.3. Application to TCGA Breast Cancer Data

We evaluated the performance of asmbPLS-DA and the other competitive methods using the TCGA breast cancer dataset, which included two types of omics data: gene expression and miRNA expression. The dataset consisted of 949 samples, including 854 tumor samples and 95 normal samples. The data were downloaded from https://portal.gdc.cancer.gov/repository. We used the tissue type (tumor vs. normal) as the binary outcome to assess the methods’ performance. Additionally, we removed samples without pathologic stage information from the 854 tumor samples, resulting in 127 samples with stage I breast cancer, 491 samples with stage II breast cancer, and 221 samples with stage III or stage IV breast cancer. We used the pathologic stage as the multiclass outcome (stage I vs. stage II vs. stage III + stage IV) for evaluation. The pre-processed procedures, including outlier removal, data normalization, and filtering, were implemented for the two omics data, resulting in a final dataset of 45,371 genes and 1410 miRNAs, with a feature ratio of approximately 32:1. For the pathologic stage outcome, we have also included one more omics data, i.e., protein expression, to further test the performance of methods. After removing samples and features with missing data, we obtained 45,371 genes and 1410 miRNAs, as well as 457 proteins for 822 tumor samples. Of these samples, 125 were in the stage I group, 480 were in the stage II group, and 217 were in the stage III or stage IV group.

We included three PLS components for PLS-based methods and performed 5-fold cross-validation with 10 repetitions for asmbPLS-DA, sPLS-DA, and IPF-Lasso for parameter tuning. DIABLO was not included due to its long computation time. For asmbPLS-DA, we used pre-determined quantile combinations of $quantile^{gene}$ = {0.999, 0.9992, 0.9994, 0.9996, 0.9998}, $quantile^{miRNA}$ = {0.96, 0.97, 0.98, 0.99, 0.995}, and $quantile^{protein}$ = {0.95, 0.96, 0.97, 0.98, 0.99}. We divided the dataset into five equal groups and used each group in turn as the test set. We calculated the overall accuracy and balanced accuracy based on the results of the classification for the test set.

#### 3.3.1. Binary Outcome (Tissue Type)

First, we used tissue type as the outcome to test the performance of different methods on the binary outcome. For asmbPLS-DA methods, for simplicity, we only used asmbPLS-DA with the decision rule of fixed cutoff for both feature selection and classification. Additionally, we utilized the features selected by asmbPLS-DA in both LDA and RF for the purpose of classification. 

Regarding feature selection, forty-six genes and eight miRNAs were selected by asmbPLS-DA; thirty genes and zero miRNAs were selected by sPLS-DA; thirty-six genes and zero miRNAs were selected by L1-regularized logistic regression; twenty-six genes and fifteen miRNAs were selected by IPF-Lasso. Since sPLS-DA and L1-regularized logistic regression do not consider group structure, no miRNA was collected by these methods, which is expected given the relatively small number of miRNAs compared to genes. In addition, due to the limitation of the Lasso method, the features selected by L1-regularized logistic regression and IPF-Lasso show relatively lower correlations compared to the features selected by asmbPLS-DA and sPLS-DA (Figure 5), indicating the effectiveness of PLS-based methods for selecting features with strong correlation. 

Table 3 lists the top five features with the largest absolute coefficients selected by each method for different blocks, along with corresponding references supporting the association between the features and breast cancer. Interestingly, asmbPLS-DA and sPLS-DA share the top list for the gene block, with VEGF-D, ADAMTS5, ABCA10, ADAM33, and PAMR1 being identified as the most important genes for breast cancer, which have been previously reported in the literature [20,21,22,23,24,25]. However, the top genes selected by L1-regularized logistic regression and IPF-Lasso differed from those identified by asmbPLS-DA and sPLS-DA, and not all of them were found to be related to breast cancer, with limited supporting literature. As for miRNAs, all of the miRNAs selected by asmbPLS-DA have been found to be associated with breast cancer in previous studies [26,27,28,29]. In contrast, the top miRNAs selected by IPF-Lasso, such as miR-6883 and miR-5191, were associated with colorectal cancer [30,31], and there was no literature to support the effects of miR-6717 and miR-548o-2 on breast cancer. As we expected, neither sPLS-DA nor L1-regularized logistic regression selected any miRNAs for breast cancer.

Table 4 lists the results of the classification for different methods, with tissue type as the outcome. For asmbPLS-DA, the quantile combination selected from the CV was used for model fitting and classification. All the methods demonstrate satisfactory classification performance in terms of overall accuracy. However, sPLS-DA exhibited the worst performance, despite performing nearly the best on the simulation datasets. Notably, 28 samples were classified as NA by sPLS-DA, which may be attributed to the unbalanced group or the excess zero values in the real data that the corresponding R function cannot handle. Among all the methods, asmbPLS-DA with RF showed the highest overall accuracy, while asmbPLS-DA with LDA exhibited the best performance in terms of balanced accuracy. L1-regularized logistic regression and IPF-Lasso exhibited high recall for the group with a larger sample size, but their performance was weaker for the smaller group.

#### 3.3.2. Multiclass Outcome (Pathologic Stage)

To further evaluate the methods, we used the pathologic stage as our multiclass outcome with more balanced group sizes of 127, 491, and 227 samples in stage I, stage II, and stage III + stage IV, respectively. Similarly, we applied asmbPLS-DA with the decision rule of Max Y for both feature selection and classification and used the selected features in LDA and RF for classification.

The results showed that asmbPLS-DA selected forty-six genes and fifteen miRNAs, sPLS-DA selected one hundred genes but no miRNA, and L1-regularized logistic regression selected forty-seven genes and one miRNA. The top five features from both blocks selected by each method are listed in Table 5. In contrast to the binary outcome results, asmbPLS-DA and sPLS-DA only shared one gene in the top list, i.e., DIRAS1, which has decreased expression in most breast cancers [38]. Among the other four genes selected by asmbPLS-DA, STOM, MMGT1, and DIAPH3 are associated with breast cancer progression [39,40,41], while FAM72B is related to lung adenocarcinoma [42]. In contrast, sPLS-DA selected RHBDF2 and CCL14 related to breast cancer [43,44], while ERG and FASTKD1 are associated with other cancers [45,46]. The associations between the genes selected by L1-regularized multinomial logistic regression and breast cancer are not supported by the available literature. Regarding miRNAs selected by asmbPLS-DA, miR-5047 regulates the proliferation and migration of breast cancer cells [47], miR-6744 increases anoikis in breast cancer [48], and miR-4634 can be used to detect breast cancer in the early stages [49]. The only miRNA selected by L1-regularized multinomial logistic regression, i.e., miR-548ag-1, is not related to breast cancer. Similarly, no miRNA was selected by sPLS-DA.

The classification results using the pathologic stage as the outcome are listed in Table 6. We found that L1-regularized multinomial logistic regression performs the best, followed by asmbPLS-DA with RF, asmbPLS-DA with LDA, sPLS-DA, and asmbPLS-DA with Max Y. Furthermore, asmbPLS-DA with LDA was noted to show the highest balanced accuracy, followed by asmbPLS-DA with Max Y, asmbPLS-DA with RF, L1-regularized multinomial logistic regression, and sPLS-DA. However, all the classification methods were found to show poor performance in classifying groups of smaller size, with L1-regularized multinomial logistic regression and asmbPLS-DA with RF nearly classifying all the samples from the stage II group. This observation is consistent with the results obtained from the real data analysis using the binary outcome.

To test the performance of the methods with more than two blocks, we added protein expression data to the analysis for the pathologic stage outcome. asmbPLS-DA identified twenty-eight genes, eight miRNAs, and fourteen proteins as the most important features. sPLS-DA identified 70 genes as the most important features but did not select any miRNAs or proteins. L1-regularized logistic regression selected sixty genes, two miRNAs, and four proteins. Notably, asmbPLS-DA selected nearly the same top genes and miRNAs as when using only two omics data (as shown in Table 7). However, sPLS-DA identified two other genes, CLEC14A and LDB2, which have been associated with breast cancer [50,51]. The genes selected by L1-regularized logistic regression did not have strong evidence supporting their relevance to breast cancer. The top miRNAs selected were similar to those from the previous results. For the protein block, the corresponding genes related to the top five proteins selected by asmbPLS-DA are all associated with breast cancer [52,53,54,55,56], while two out of the three proteins selected by L1-regularized logistic regression are related to breast cancer [57,58]. However, including the protein block did not significantly improve the classification performance of any of the methods (Appendix A).

## 4. Conclusions

In this paper, we extended our previous work, asmbPLS, to a new method called asmbPLS-DA. This new method considers both binary and multiclass categorical outcomes. The aim of asmbPLS-DA is to identify the most predictive features in diverse types of omics data and then use the selected features for classification. This helps to improve our understanding of the molecular mechanisms behind diseases. To achieve this goal, we have included different decision rules and several vote functions. Additionally, we have combined asmbPLS-DA with other classification algorithms for further testing. The proposed method is implemented in the R package *asmbPLS* available on our GitHub (https://github.com/RunzhiZ/asmbPLS), and it will soon be uploaded to CRAN.

Simulation studies have shown that asmbPLS-DA has very satisfactory performances regarding feature selection, especially for multiclass outcomes and cases where different omics data vary greatly in the number of features. In methods that do not consider the group structure, such as sPLS-DA, L1-regularized logistic regression, and L1-regularized multinomial logistic regression, the feature selection can be dominated by the block with the greatest number of features, ignoring information from smaller blocks. However, due to the nature of asmbPLS-DA, it can select the predictive features from all the blocks included, even if these features are highly correlated. The feature selection performance of asmbPLS-DA is stable regardless of the scenario setting. In real data, the top features selected by asmbPLS-DA have been validated by the literature, further indicating the power of feature selection by asmbPLS-DA. 

In the simulation studies, asmbPLS-DA performs worse overall for classification compared to the other methods, except for DIABLO. However, RF with the features selected by asmbPLS-DA shows similar performance to the best in the scenarios where all the relevant features are highly correlated, which is quite common for multi-omics data. This indicates the significance of the selected features. Moreover, as the noise increases, the performances of asmbPLS-DA methods get closer to the best. In the real data, we found that asmbPLS-DA with LDA shows the best classification when considering both the overall accuracy and the balanced accuracy, while asmbPLS-DA with RF shows good performance regarding the overall accuracy only.

The current version of asmbPLS-DA does not include the pre-processing of missing values. Additionally, if there are no relevant features in the predictor blocks, classification can be problematic for the method since asmbPLS-DA will always select some features for each block, resulting in non-relevant features being selected.

In summary, we have shown that asmbPLS-DA is a useful tool for multi-omics analysis. We believe that asmbPLS-DA provides a valuable alternative for both feature selection and classification in binary and multiclass problems.

## Figures and Tables

**Figure 1 genes-14-00961-f001:**
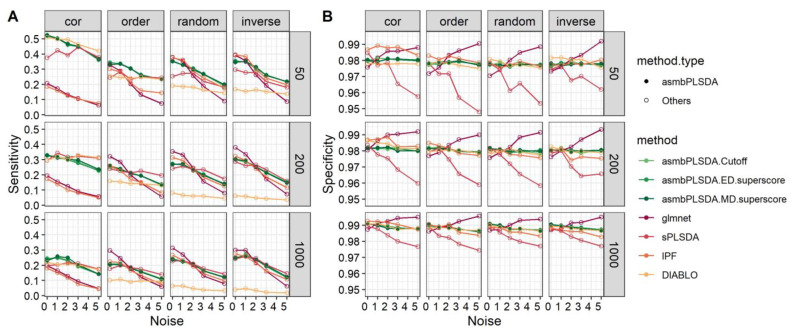
The overall sensitivity and specificity of different methods for simulation data with the binary outcome. (**A**) sensitivity; (**B**) specificity.

**Figure 2 genes-14-00961-f002:**
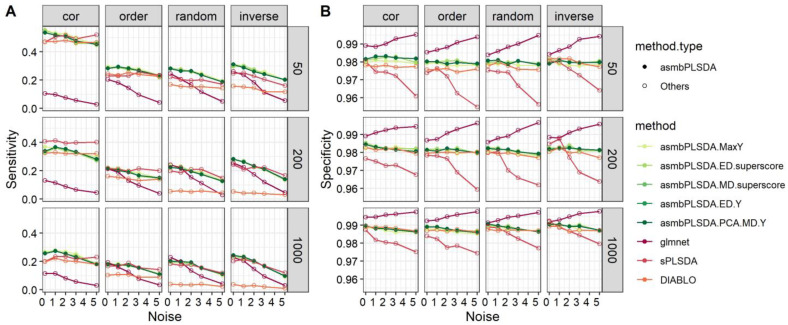
The overall sensitivity and specificity of different methods for simulation data with the multiclass outcome. (**A**) sensitivity; (**B**) specificity.

**Figure 3 genes-14-00961-f003:**
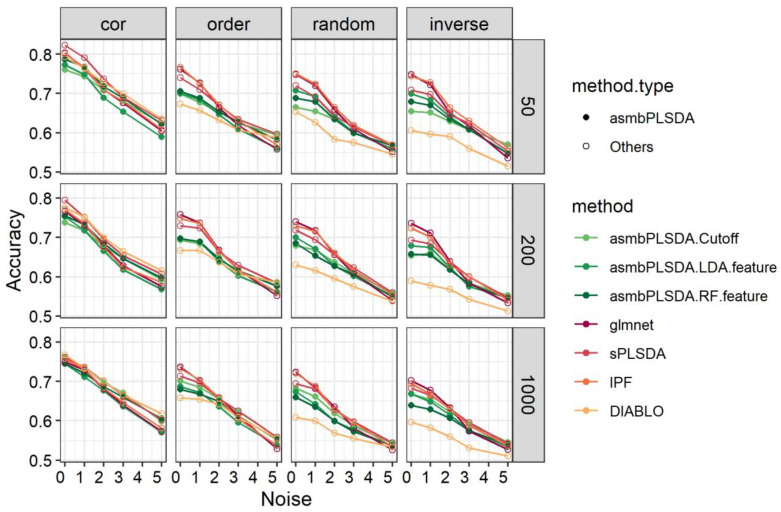
The classification results for simulation data with the binary outcome.

**Figure 4 genes-14-00961-f004:**
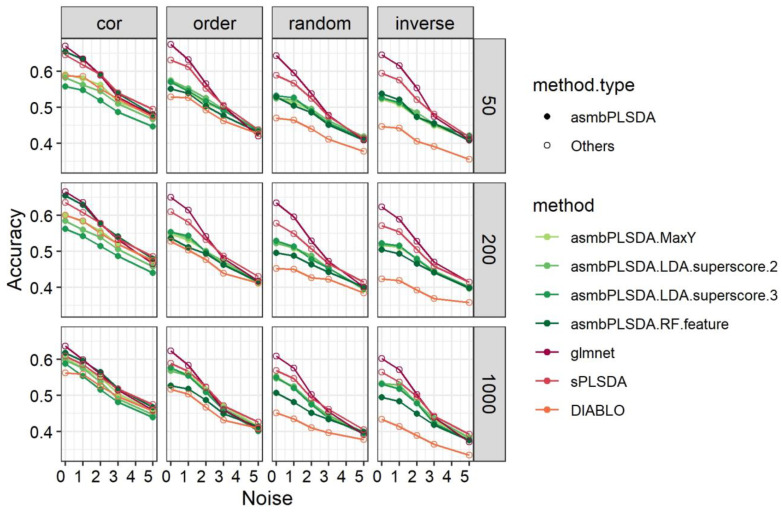
The classification results for simulation data with the multiclass outcome.

**Figure 5 genes-14-00961-f005:**
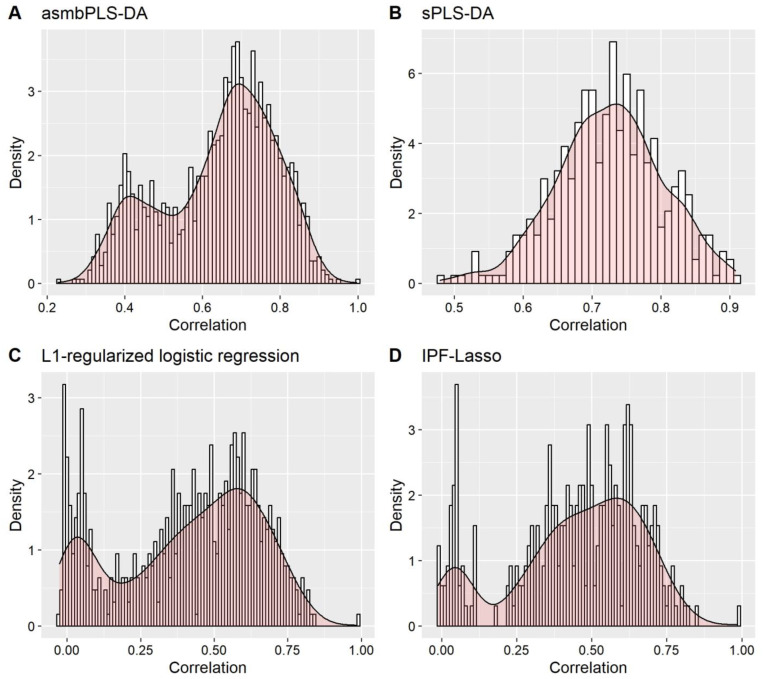
The correlations between the features selected by different methods for real data with the binary outcome.

**Table 1 genes-14-00961-t001:** Descriptions of different decision rules. All Y and $t^{super}$ in the table indicate the predicted values. √ and × indicate if the decision rule can be applied to the corresponding object.

Decision Rule	Object	Binary	Multiclass	Definition
Fixed Cutoff	Y	√	×	Y-estimates over 0.5 are assigned to 1, while Y-estimates under 0.5 are assigned to 0.
Max Y	Y	×	√	The sample is assigned to the group with the maximum Y-estimate.
Euclidean Distance (ED)	$t^{super}$	√	√	The sample is assigned to the group with the shortest ED using $t^{super}$ or Y.
	Y	×	√	
Mahalanobis Distance (MD)	$t^{super}$	√	√	The sample is assigned to the group with the shortest MD using $t^{super}$.
Principal Components Analysis (PCA) + MD	Y	×	√	PCA is applied on Y. The sample is assigned to the group with the shortest MD using the first G−1 principal components of Y.

**Table 2 genes-14-00961-t002:** Average computation time (in seconds) for different methods.

p	Outcome	asmbPLS-DA	sPLS-DA	Glmnet	IPF-Lasso	DIABLO
50	binary	8.972	17.547	0.186	8.677	225.285
200	binary	10.017	18.916	0.203	9.600	223.087
1000	binary	23.956	24.712	0.270	12.513	273.367
50	multiclass	48.966	34.843	0.861	-	231.654
200	multiclass	59.892	40.006	0.961	-	244.613
1000	multiclass	108.989	45.270	1.079	-	267.646

**Table 3 genes-14-00961-t003:** The top five features selected by different methods for the gene and the miRNA blocks, respectively, using tissue type as the outcome. The symbol * indicates the evidence of the association between the feature and other types of cancer other than breast cancer.

Block	Feature		Ranking				Reference
	ID	Gene	asmbPLS-DA	sPLS-DA	L1-Regularized Logistic Regression	IPF-Lasso	
Gene	ENSG00000165197	VEGFD	1	1	-	-	[24,25]
	ENSG00000154736	ADAMTS5	2	2	-	-	[22]
	ENSG00000154263	ABCA10	3	3	-	-	[20]
	ENSG00000149451	ADAM33	4	4	-	-	[21]
	ENSG00000149090	PAMR1	5	5		-	[23]
	ENSG00000148204	CRB2	-	-	1	3	[32]
	ENSG00000228868	MYLKP1	-	-	2	2	[33]
	ENSG00000229246	LINC00377	-	-	3	-	[34]
	ENSG00000197891	SLC22A12	-	-	4	-	-
	ENSG00000240654	C1QTNF9	-	-	5	-	[35]
	ENSG00000257766	LOC105369946	-	-	-	1	-
	ENSG00000226005	LINC02660	-	-	-	4	-
	ENSG00000236036	LINC00445	-	-	-	5	[36]
miRNA	hsa-miR-139		1	-	-	-	[28]
	hsa-miR-145		2	-	-	-	[29]
	hsa-miR-10b		3	-	-	-	[26]
	hsa-miR-125b-1		4	-	-	-	[27]
	hsa-miR-125b-2		5	-	-	-	[27]
	hsa-miR-6883 *		-	-	-	1	-
	hsa-miR-5191 *		-	-	-	2	-
	hsa-miR-6717		-	-	-	3	-
	hsa-miR-3912		-	-	-	4	[37]
	hsa-miR-548o-2		-	-	-	5	-

**Table 4 genes-14-00961-t004:** Comparison of the classification performance for different methods using the real data with the binary outcome.

Method	Overall Accuracy	Recall for Normal Group	Recall for Tumor Group	Balanced Accuracy
asmbPLS-DA with fixed cutoff	0.9852	0.9579	0.9883	0.9731
asmbPLS-DA with LDA	0.9916	0.9684	0.9941	0.9813
asmbPLS-DA with RF	0.9926	0.9474	0.9977	0.9725
sPLS-DA	0.9642	0.9474	0.9660	0.9567
L1-regularized logistic regression	0.9895	0.9263	0.9965	0.9614
IPF-Lasso	0.9905	0.9263	0.9977	0.9620

**Table 5 genes-14-00961-t005:** The top five features selected by different methods for the gene and the miRNA blocks, respectively, using the pathologic stage as the outcome. The symbol * indicates the evidence of the association between the feature and other types of cancer other than breast cancer.

Block	Feature		Ranking			Reference
	ID	Gene	asmbPLS-DA	sPLS-DA	L1-Regularized Multinomial Logistic Regression	
Gene	ENSG00000176490	DIRAS1	1	1	-	[38]
	ENSG00000148175	STOM	2	-	-	[41]
	ENSG00000169446	MMGT1	3	-	-	[40]
	ENSG00000139734	DIAPH3	4	-	-	[39]
	ENSG00000188610	FAM72B *	5	-	-	-
	ENSG00000129667	RHBDF2	-	2	-	[44]
	ENSG00000276409	CCL14	-	3	-	[43]
	ENSG00000157554	ERG *	-	4	-	-
	ENSG00000138399	FASTKD1 *	-	5	-	-
	ENSG00000250115	AK3P2	-	-	1	-
	ENSG00000189372	-	-	-	2	-
	ENSG00000218233	NEPNP	-	-	3	-
	ENSG00000267583	ZNF24TR			4	-
	ENSG00000260017	-			5	-
miRNA	hsa-miR-6503		1	-	-	-
	hsa-miR-5047		2	-	-	[47]
	hsa-miR-6744		3	-	-	[48]
	hsa-miR-4439		4	-	-	-
	hsa-miR-4634		5	-	-	[49]
	hsa-miR-548ag-1		-	-	1	-

**Table 6 genes-14-00961-t006:** Comparison of the classification performance for different methods using the real data with the multiclass outcome.

Method	Overall Accuracy	Recall for Stage I Group	Recall for Stage II Group	Recall for Stage III + Stage IV Group	Balanced Accuracy
asmbPLS-DA with Max Y	0.4708	0.2283	0.6334	0.2489	0.3702
asmbPLS-DA with LDA	0.5650	0.1811	0.8432	0.1674	0.3972
asmbPLS-DA with RF	0.5733	0.0079	0.9552	0.0498	0.3376
sPLS-DA	0.5292	0.0000	0.8554	0.1086	0.3213
L1-regularized multinomial logistic regression	0.5805	0.0000	0.9898	0.0045	0.3314

**Table 7 genes-14-00961-t007:** The top five features selected by different methods for the gene, miRNA, and protein blocks, respectively, using the pathologic stage as the outcome. The symbol * indicates the evidence of the association between the feature and other types of cancer other than breast cancer.

Block	Feature		Ranking			Reference
	ID	Gene	asmbPLS-DA	sPLS-DA	L1-Regularized Multinomial Logistic Regression	
Gene	ENSG00000176490	DIRAS1	1	2	-	[38]
	ENSG00000169446	MMGT1	2	-	-	[40]
	ENSG00000148175	STOM	3	-	-	[41]
	ENSG00000188610	FAM72B *	4	-	-	-
	ENSG00000139734	DIAPH3	5	-	-	[39]
	ENSG00000129667	RHBDF2	-	1	-	[44]
	ENSG00000176435	CLEC14A	-	3	-	[50]
	ENSG00000157554	ERG *	-	4	-	-
	ENSG00000169744	LDB2	-	5	-	[51]
	ENSG00000250115	AK3P2	-	-	1	-
	ENSG00000213539	YBX1P6	-	-	2	-
	ENSG00000189372	-	-	-	3	-
	ENSG00000265684	RN7SL378P	-	-	4	-
	ENSG00000272226	-	-	-	5	-
miRNA	hsa-miR-6503		1	-	-	-
	hsa-miR-4439		2	-	-	-
	hsa-miR-6744		3	-	2	[48]
	hsa-miR-548ag-1		4	-	1	-
	hsa-miR-5047		5	-	-	[47]
Protein	AGID02152	FOXM1	1	-	-	[56]
	AGID00255	SLFN11	2	-	4	[53]
	AGID00391	AURKA	3	-	-	[55]
	AGID00293	CDK1	4	-	-	[54]
	AGID00016	CAV1	5	-	-	[52]
	AGID00098	MAPK11	-	-	1	[58]
	AGID00062	RPS6	-	-	2	[57]
	AGID00295	H2BC3	-	-	3	-

## Data Availability

The real data were obtained from the Genomic Data Commons Data Portal (https://portal.gdc.cancer.gov/repository) (accessed on 9 January 2023), where the cancer datasets have been gathered by the TCGA research network. We downloaded the gene, miRNA, and protein, as well as clinical datasets, from the TCGA-BRCA project.

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
