# Peer review of "Adaptive Sparse Multi-Block PLS Discriminant Analysis: An Integrative Method for Identifying Key Biomarkers from Multi-Omics Data"

_genes, 2023, doi:10.3390/genes14050961_

Round 1

Reviewer 1 Report

#Minor

1) For section 2.1.1 Model Building (line 94), please include the symbol for the end of jth PLS component (j = 1,2, …, J?).

2) Please reorganize and include the CV procedure for the asmbPLS-DA algorithm in the Table2 into the text in the section 2.2.3 or the supplementary section.

3) For section 2.2.4 Vote Algorithm (line 175), please use the consistent style for the list of vote algorithms. i.e. 1), 2), 3)..

4) For Figure 1-5, please use the consistent style for the axis label (capitalize the first character).

5) Please include a tutorial for the asmbPLS package that utilize the provided data on the GitHub repository, as the authors states that the package includes visualization functions that can assist with result interpretation.

Author Response

Thank you for your thoughtful and constructive feedback on our manuscript. Your comments have helped us significantly improve our paper. We have responded to each of your comments below.

Minor comments:

  • For section 2.1.1 Model Building (line 94), please include the symbol for the end of jth PLS component (j = 1,2, …, J?).

Author's response: Thank you for pointing this our. We have updated the symbol in the revised manuscript. (line 94).

  • Please reorganize and include the CV procedure for the asmbPLS-DA algorithm in the Table2 into the text in the section 2.2.3 or the supplementary section.

Author's response: Thank you for your valuable suggestion . We have revised our manuscript in the section 2.2.3. The Table 2 is now removed and the CV procedure is included in the text. (line 160 – 178 in the revised manuscript).

  • For section 2.2.4 Vote Algorithm (line 175), please use the consistent style for the list of vote algorithms. i.e. 1), 2), 3)..

Author's response: We apologize for the typo here. We have addressed this in the revised manuscript (line 194 in the revised manuscript).

  • For Figure 1-5, please use the consistent style for the axis label (capitalize the first character).

Author's response: Thank you for bringing this to our notice. We have updated our figures to have the consistent type for the axis label. In addition to this, we have also updated the color for each method in Figures 1-4 to make them more readable. Additionally, we would like to bring to your attention that we have made an update to Figure 5.A. Upon further review, we found that the original figure was based on outdated results due to a different seed set used during the analysis. We have since corrected this by updating the figure to reflect the most current results. It's important to note that this update will not affect any of the discussion or conclusions made in the manuscript.

  • Please include a tutorial for the asmbPLS package that utilize the provided data on the GitHub repository, as the authors states that the package includes visualization functions that can assist with result interpretation.

Author's response: Thank you for your valuable suggestion. Now we have updated GitHub page and the tutorial is available now. Also, we have included vignettes in our R package, when people install the R package with vignettes built, they will also be able to find the tutorial (vignettes) in the R package.

Reviewer 2 Report

The we present an Adaptive Sparse Multi-omics Partial Least Square Discriminant Analysis for identifying key biomarkers from multi-omics data. The study is interesting. It can be a valuable tool for multi-omics research. Overall, the structure of the text (objectives, results and discussion) is organized in a way that allow the logical understanding of the scientific reasoning of the authors.

Author Response

The we present an Adaptive Sparse Multi-omics Partial Least Square Discriminant Analysis for identifying key biomarkers from multi-omics data. The study is interesting. It can be a valuable tool for multi-omics research. Overall, the structure of the text (objectives, results and discussion) is organized in a way that allow the logical understanding of the scientific reasoning of the authors.

Author's response: We want to thank the reviewer for the valuable comments and the overall evaluation. We have improved our manuscript to make it a better version. We would like to bring to your attention that we have made an update to Figure 5.A. Upon further review, we found that the original figure was based on outdated results due to a different seed set used during the analysis. We have since corrected this by updating the figure to reflect the most current results. It's important to note that this update will not affect any of the discussion or conclusions made in the manuscript.

Reviewer 3 Report

The proposed article by Zhang and Datta, refers to a new package and model, to identify the most relevant features across different types of omics data. While revision this manuscript I found several issues that needs to be deeply addressed before a more detailed revision.

Major issues:

- In this article it is state that asmb-PLS-DA is build on top of asmbPLS model. Although, this is the first publication of asmbPLS model. I believe that the authors should clarify this from the beginning (on introduction it seems that asmbPLS is a model already community validated). asmbPLS is not a "previous" model if it is made known here.

- Authors are advised to number the equations. It is almost impossible to review this paper without that numbering system.

- Several software misses version. Please revise the entire manuscript and add missing data on methods. For software where version is not available, please include download date. Experiment replication is impossible without this and the article should be rejected on this ground.

- Information regarding real data used is missing. Authors explained how data was downloaded and which database used, but not in the methods, and do not provide deep information on the accessions used. Experiment replication is impossible without this and the article should be rejected on this ground.

- Methods section are incomplete. On line 230[-231], authors say that «(...) And we used posdefify function from the R package sfsmisc [17] (...)». Methods section does not refer to this. This means that results are by non refereed methods. Experiment replication is impossible without this and the article should be rejected on this ground.

Please note: Considering the major issued refereed above, I decided to recommend the rejection of this manuscript and to stop my revision. Therefore, the authors are advised to spend more time and improve the proposed article before submitting it to a second round of revisions. Nevertheless, I hope that the revision and comments provided so far may help the authors to improve their manuscript.

Minor issues:

- Keywords: This field is normally used for article indexation. With
this in mind I would recommend the change between asmb-PLS-DA and
PLS-DA.

- Line 83: Please replace "are" with "were" on «Different decision rules [are/were] included for the purpose».

- Line 126[-127]: On «Usually, for the binary outcome, 0.5 is used as the cutoff for the group decision when we have equal sample sizes in two groups.», the authors should justify their claims or cite corresponding articles.

- Line 175: Please replace "a)" to "1)" on «(...) help the discrimination: [a/1]) Unweighted vote,(...) 2) (...)»

- Results: Authors are advised to rename "Results" to "Results and Discussion". Also...

- Discussion: Authors are advised to rename "Discussion" to "Conclusion"

- Table 3: This table may be improved to increase readability: 1) if n, q and qh are the same, why put them on the table? Please refer to this on the caption. 2) The caption should refer to p and ph meanings. Readers should not require to refer to the main text to fast-scan tables and figures.

Author Response

We wish to express our strong appreciation for the insightful comments on our paper. We have responded to each of your comments below.

Minor issues:

  • In this article it is state that asmb-PLS-DA is build on top of asmbPLS model. Although, this is the first publication of asmbPLS model. I believe that the authors should clarify this from the beginning (on introduction it seems that asmbPLS is a model already community validated). asmbPLS is not a "previous" model if it is made known here.

Author's response: Thank you for pointing this out. We want to explain a little bit about the situation. We have submitted our asmbPLS paper to another journal last year, and we have received quite positive comments with only minor revision required and we submitted the revision the end of last year. However, it took longer time than we expected and we haven’t heard further decision yet. To address this, we have put our asmbPLS paper on bioRxiv (https://www.biorxiv.org/content/10.1101/2023.04.03.535442v1).

  • Authors are advised to number the equations. It is almost impossible to review this paper without that numbering system.

Author's response: Thank you for the valuable suggestion. We have revised the manuscript and numbered the equations.

  • Several software misses version. Please revise the entire manuscript and add missing data on methods. For software where version is not available, please include download date. Experiment replication is impossible without this and the article should be rejected on this ground.

Author's response: Thank you for pointing this out. Now we have updated the method section to include the version information for all the software used. (line 209 – 228 in the revised manuscript)

  • Information regarding real data used is missing. Authors explained how data was downloaded and which database used, but not in the methods, and do not provide deep information on the accessions used. Experiment replication is impossible without this and the article should be rejected on this ground.

Author's response: Thank you for bringing this to our notice. Now we have updated the method section to include the detailed information about the real data used. (line 213 – 220 in the revised manuscript)

  • Methods section are incomplete. On line 230[-231], authors say that «(...) And we used posdefify function from the R package sfsmisc [17] (...)». Methods section does not refer to this. This means that results are by non refereed methods. Experiment replication is impossible without this and the article should be rejected on this ground.

Author's response: Thank you for pointing this out. Now we have updated the method section to to address this missing. (line 221 – 222 in the revised manuscript)

Minor issues:

  • Keywords: This field is normally used for article indexation. With this in mind I would recommend the change between asmb-PLS-DA and PLS-DA.

Author's response: Thank you for the valuable suggestion. We agree with your point of view and we have revised it in the manuscript.

  • Line 83: Please replace "are" with "were" on «Different decision rules [are/were] included for the purpose».

Author's response: Thank you for pointing this out. We have replaced “are” with “were” in the manuscript. (line 83 in the revised manuscript)

  • Line 126[-127]: On «Usually, for the binary outcome, 0.5 is used as the cutoff for the group decision when we have equal sample sizes in two groups.», the authors should justify their claims or cite corresponding articles.

Author's response: Thank you for pointing this out. We have revised the manuscript and added the explanation for the value of 0.5 used here. For binary outcome, we use 1 and 0 to indicate different group, the average of two group indicators (1+0)/2 is then used as the cutoff. (line 127 -128 in the revised manuscript).

  • Line 175: Please replace "a)" to "1)" on «(...) help the discrimination: [a/1]) Unweighted vote,(...) 2) (...)»

Author's response: We apologize for the typo here. We have addressed this in the revised manuscript (line 194 in the revised manuscript).

  • Results: Authors are advised to rename "Results" to "Results and Discussion". Also...

Author's response: Thank you for the valuable suggestion. We agree with your point of view and we have revised it in the manuscript.

  • Results: Authors are advised to rename "Results" to "Results and Discussion". Also... Discussion: Authors are advised to rename "Discussion" to "Conclusion"

Author's response: Thank you for the valuable suggestion. We agree with your point of view and we have revised it in the manuscript.

  • Table 3: This table may be improved to increase readability: 1) if n, q and qh are the same, why put them on the table? Please refer to this on the caption. 2) The caption should refer to p and ph meanings. Readers should not require to refer to the main text to fast-scan tables and figures.

Author's response: Thank you for the valuable suggestion. We have decided to remove Table 3 in the manuscript and the details of the parameters used are now added in the main text. (line 262 - 265 in the revised manuscript)